# Immune Checkpoint Inhibitors-Related Myocarditis: A Review of Reported Clinical Cases

**DOI:** 10.3390/diagnostics13071243

**Published:** 2023-03-25

**Authors:** Liudmila Zotova

**Affiliations:** Department of Hospital Therapy with a Course of Medical and Social Expertise, Federal State Budgetary Educational Institution of Higher Education “Ryazan State Medical University named after Academician I.P. Pavlova” of the Ministry of Health of the Russian Federation, Ryazan 390026, Russia; dr.zotova@gmail.com

**Keywords:** immune checkpoint inhibitors, myocarditis, cardiotoxicity, immunotherapy-related adverse reactions, case report

## Abstract

Myocarditis associated with the use of immune checkpoint inhibitors (ICI) is a rare manifestation of their cardiotoxicity, but is characterized by a high mortality rate. A literature search was conducted using PubMed using keywords, which resulted in the selection of 679 scientific works, from which 160 articles that described 244 clinical cases were selected. The median age of the patients was 67 years (IQR, 60–74). The median time from the start of ICI therapy to the development of the first adverse symptoms was 21 days (IQR, 14–38.3). In 37% of cases, myocarditis developed after the first administration of ICI. Cardiac symptoms were present in 47.1% of cases, neuromuscular symptoms in 30.3%, and other symptoms in 12.6%, while myocarditis was asymptomatic in 10.1% of cases. New changes in the electrocardiograms were detected in 85.1% of patients compared to the initial data. A high incidence of complete atrioventricular block (25.4%), right bundle branch block (18.4%), ventricular tachycardia (13%), and sinus tachycardia (12%) were noted. In 97% of the cases, the patients received prednisolone or methylprednisolone therapy. When using ICI, special attention should be paid to the early detection of possible cardiotoxicity by analyzing the condition and function of the myocardium before treatment and its dynamics.

## 1. Introduction

Statistical data show that over the past few decades, the incidence of malignant neoplasms has been steadily increasing, which can be explained by various factors, including population aging, particularly in countries with a high level of economic development, improvements in diagnostic capabilities, and advances in the treatment of oncological diseases, which in turn is reflected in the reduction in mortality rates [1]. During this time, a number of new groups of drugs have been developed and implemented in clinical practice in oncology, and immunotherapy has been actively developed, including monoclonal antibodies, non-specific cytokines, anti-cancer vaccines, and immune checkpoint inhibitors (ICI), which have made a breakthrough in the treatment of a whole range of oncological diseases. The first ICI approved by the Food and Drug Administration (FDA) was the drug ipilimumab (cytotoxic T-lymphocyte associated protein 4 (CTLA-4) inhibitor), which was approved for the treatment of melanoma in 2011 [2]. The next drug in this group to be approved by the FDA was nivolumab (programmed cell death ligand 1 (PD-1) molecular inhibitor) in 2014. Over the next few years, new programmed cell death protein 1 (PD-1) or its ligand, PD-L1, receptor inhibitors were approved all over the world [3,4]. The spectrum of indications for the use of this group of immune drugs in oncology practice is very wide (Table 1).

The mechanism of action of these drugs involves blocking various checkpoint proteins, including the following: CTLA-4 (cytotoxic T-lymphocyte-associated protein 4), PD-1 (programmed cell death protein 1), PD-L1 (programmed cell death ligand 1), LAG-3 (lymphocyte-activation gene 3).

Both CTLA-4 and PD-1, along with its partner PD-L1, are checkpoint proteins that can be inhibited by drugs. Other immune checkpoint drugs target PD-1 and PD-L1, which are partners in the immune system. Certain tumors overproduce PD-L1, which can inhibit the T cell response. Using this process, tumor cells increase the expression of PD-L1 to avoid recognition and destruction by the immune system. ICI block these immune checkpoints, “turning on” the cellular immune response against tumor cells. The mechanisms of action of these antibodies complement each other, which is why they are often used in combination. Other immune checkpoints that are currently being studied for their therapeutic potential include T cell immunoglobulin and mucin-containing protein 3 (TIM-3) [11], lymphocyte-activated gene-3 (LAG-3) [12], T cell immunoreceptor with Ig and ITIM domains (TIGIT) [13], B and T lymphocyte attenuator (BTLA) [14], V-domain Ig suppressor of T cell activation (VISTA, also known as PD-1 homologue, or PD-1H) [15], and others.

Due to their direct effect on the immune system, this class of drugs has specific immune-related adverse events (irAEs). The widespread use of ICI has led to an increase in the frequency of occurrence of specific toxicities. According to the data of various authors, the frequency of irAEs is around 30% with the use of CTLA-4 inhibitors, 20% with anti-PD-1 inhibitors, and 55% in patients receiving a combination of drugs [16]. ICI can have different toxicity profiles, such as dermatological, renal, gastrointestinal, cardiovascular, endocrine, neurological, ocular, pulmonary, musculoskeletal and hematological profiles [17].

IrAEs occur in 70–90% of patients receiving ICI, with severe irAEs occurring in 10–15% of patients [18]; these reactions result in a fatal outcome in 1.3% of patients [19]. IrAEs often occur within 1 year after the treatment [20], but the risk of developing any irAE increases over time [21].

Accumulated data formed the basis for recommendations for the diagnosis and treatment of irAEs [22,23,24]. When assessing the severity of IrAEs, general terminology criteria for adverse events are used, as recommended by the European Society of Medical Oncology and the American Society of Clinical Oncology (Table 2) [25,26].

Cardiotoxicity associated with ICI is relatively rare. However, this type of IrAE often leads to a fatal outcome [27]. Various manifestations of cardiotoxicity are possible, including myocarditis, arrhythmia, asequence, acute coronary syndrome, heart failure, and pericarditis [28]. The most common type of cardiotoxicity associated with ICI is myocarditis. The frequency of ICI-associated myocarditis is known to be between 0.09% and 2.4% [29,30,31]. However, the mortality rate is very high, at approximately 27–60% [32,33]. In 46% of cases, major adverse cardiac events (MACE) occur, including stroke, myocardial infarction, cardiovascular death, cardiogenic shock, cardiac arrest and hospitalization for severe heart failure [30].

The mechanism of cardiotoxicity is not fully studied, but the main cause is attributed to CD4+ T-cell-mediated inflammation [34]. The most common cytokines produced are tumor necrosis factor-alpha (TNF-α), granzyme B, and interferon gamma [35]. In addition, the expression of inflammatory transcription factors NLR family pyrin domain containing 3 (NLRP3), myeloid differentiation primary response gene 88 (MyD88) and p65/NF-κB is increased in cardiomyocytes [36]. It has been established that ICI does not induce the apoptosis of cardiomyocytes, meaning it does not have a direct cytotoxic effect [37].

Wang et al. demonstrated that fatal myocarditis developed in mice genetically predisposed the mice to systemic autoimmunity due to PD-1 deficiency, whereas in mice with genetic predisposition to autoimmunity but without PD-1 deficiency, myocarditis did not develop, suggesting that myocarditis prevention is likely to be mediated by PD-1 [38]. In a similar study, Love et al. found that the deletion of CTLA-4 on T cells also caused severe myocarditis in mice, but the absence of IL-12 prevented the proliferation of CD8+ T cells, thereby reducing the risk of developing myocarditis [39]. PD-L1, expressed in human myocardium, is involved in protecting against immune-mediated heart damage and inflammation [37].

In this article, a review of previously described cases of myocarditis caused by ICI is presented. The clinical demographic characteristics were analyzed, as well as the changes in the electrocardiograms (ECG), patient complaints, changes in laboratory parameters, data from echocardiography (EchoCG), cardiac magnetic resonance imaging (MRI), endomyocardial biopsy and performed therapy and outcomes.

## 2. Materials and Methods

The PubMed database was used to search for articles. The search was conducted using the following terms: (Checkpoint myocarditis) OR (Ipilimumab myocarditis) OR (Nivolumab myocarditis) OR (Pembrolizumab myocarditis) OR (Cemiplimab myocarditis) OR (Atezolizumab myocarditis) OR (Avelumab myocarditis) OR (Durvalumab myocarditis) or (Tremelimumab myocarditis) or (Relatlimab myocarditis). A total of 679 records were identified in the PubMed database. Subsequently, articles that were not accessible for research, articles not in English, and articles that did not contain descriptions of clinical cases of ICI-associated myocarditis were excluded. Some of the excluded data were used for the review section of our article. Additionally, the list of literature identified in the initially identified articles was manually checked to search for additional clinical cases. After that, duplicated descriptions of clinical cases that were found in different articles from the same institution were excluded. In the end, 160 articles describing 244 cases were selected [16,29,31,40,41,42,43,44,45,46,47,48,49,50,51,52,53,54,55,56,57,58,59,60,61,62,63,64,65,66,67,68,69,70,71,72,73,74,75,76,77,78,79,80,81,82,83,84,85,86,87,88,89,90,91,92,93,94,95,96,97,98,99,100,101,102,103,104,105,106,107,108,109,110,111,112,113,114,115,116,117,118,119,120,121,122,123,124,125,126,127,128,129,130,131,132,133,134,135,136,137,138,139,140,141,142,143,144,145,146,147,148,149,150,151,152,153,154,155,156,157,158,159,160,161,162,163,164,165,166,167,168,169,170,171,172,173,174,175,176,177,178,179,180,181,182,183,184,185,186,187,188,189,190,191,192,193,194,195,196,197]. The majority of the scientific works were dated 2022 and 2021.

The continuous variables are presented as medians with interquartile ranges (IQR, Q1–Q3). The categorical variables are presented as frequencies (N) and percentages (%).

## 3. Results

### 3.1. The Analyzed Patients

In total, 244 cases of ICI-associated myocarditis were identified. The median age of the patients was 67 years (IQR, 06–74) and the patients were predominantly males (143; 40.6%). The age at which ICI-associated myocarditis developed was 66 years (IQR, 56–71) for females and was lower than that for males aged 69 years (IQR, 63–75).

In the analyzed group, patients received ICI therapy for the following types of cancer: melanoma (54 cases, 22.1%), lung cancer (51 cases, 20.9%), renal cell carcinoma (26 cases, 10.7%), thymoma (16 cases, 6.6%), gastric cancer (13 cases, 5.3%), hepatocellular carcinoma (10 cases, 4.1%), urothelial carcinoma (9 cases, 3.7%), and other locations with 1–8 described cases (mesothelioma, urothelial carcinoma, breast cancer, esophageal squamous cell carcinoma, cutaneous squamous cell carcinoma, prostate adenocarcinoma, and others) and 61 cases (25%).

An analysis of the administered therapy showed that 73% of the ICI-associated myocarditis cases occurred during monotherapy with anti-PD-1 drugs; in 17.6% of cases, a combination of anti-CTLA4 and anti-PD-1 therapy was used, while cases of myocarditis during monotherapy with anti-CTLA-4 (0.8%), anti-PD-L1 (7.8%), and combined anti-PD-1/anti-PD-L1 therapy (0.8%) were much less frequent. ICI-associated myocarditis most often occurred during the use of pembrolizumab (30.7%), nivolumab (20.1%), and combined therapy with nivolumab plus ipilimumab (16.8%). Based on these data, it can be assumed that the drugs in the anti-PD-1 group, especially in combination with anti-CTLA4 drugs, have the highest cardiotoxicity among ICI. The prescription of drugs in this group may be one of the risk factors for the development of autoimmune myocarditis, although this hypothesis requires further investigation in prospective studies. The characteristics of the analyzed group of patients are presented in Table 3.

The median time from the start of immunotherapy to the development of the first adverse symptoms was 21 days (IQR, 14–38.3). In 37% of cases, myocarditis developed after the first administration of ICI. However, there were cases where the first symptoms appeared later than a year after the start of ICI (both cases in the pembrolizumab group); in addition, cases of onset after more than 6 months of use of ICI were described in the pembrolizumab, nivolumab, and nivolumab plus ipilimumab groups. The median number of days from the start of ICI varied in the following ways depending on the treatment option: pembrolizumab—26 days (IQR, 16–61), nivolumab—23 days (IQR, 14–36), nivolumab plus ipilimumab—19 days (IQR, 12–32), sintilimab—21 days (IQR, 21–37.5), camrelizumab—25 days (IQR, 14–39.5), durvalumab—34 days (IQR, 26–45.5), toripalimab—20 days (IQR, 14–29.5), atezolizumab—14 days (IQR, 4–30); tislelizumab—20 days (IQR, 11–20) (Table 4).

In six of the analyzed works, there was no description of symptoms. The manifestation of ICI-associated myocarditis is characterized by a significant heterogeneity of symptoms. Depending on the prevailing type of symptoms, patients were divided into cases with predominantly cardiac symptoms (shortness of breath, chest pain, palpitations, heart rhythm disturbances, etc.), neuromuscular symptoms (muscle weakness, double vision, drooping eyelids, etc.), and other manifestations (fever, rash, diarrhea, etc.). Cardiac symptoms bothered patients in 47.1% of cases, neuromuscular symptoms in 30.3%, and other symptoms in 12.6%. It is noteworthy that in 10.1% of cases, ICI-associated myocarditis began asymptomatic and was diagnosed based on laboratory and/or ECG and/or cardiac MRI signs.

We conducted an analysis of the ECG changes associated with the development of ICI-associated myocarditis (Table 5). In 43 cases, there was no description of ECG; therefore, the evaluation was performed based on the data of 201 cases.

In total, 85.1% of patients had new ECG changes compared to before starting ICI. Various arrhythmias, conduction disturbances, ST segment changes, T wave inversion, new Q waves, and low voltage in multiple leads were recorded. Conduction disturbances predominated, including atrioventricular block of varying degrees and bundle branch blocks. Atrioventricular block was registered in 35.4% of cases, with complete atrioventricular block being the most common type. Bundle branch block was represented by left bundle branch block (6%) and right bundle branch block (18.4%). Arrhythmias manifested as supraventricular and ventricular variants. Among the supraventricular variants, sinus tachycardia was the most common type (12%). Among the cases analyzed, 13% of patients developed ventricular tachycardia. In addition to the described changes, changes in the ST segment were registered in 12.4% of cases. The detection of such changes was the basis for differential diagnosis with acute coronary syndrome and coronary angiography was performed. In all cases, no obstructive changes were detected in the coronary arteries, and subsequent dynamics of symptoms and laboratory parameters, as well as additional investigations (MRI and endomyocardial biopsy), allowed the exclusion of acute forms of ischemic heart disease. Low voltages were observed in several leads in 4% of cases, which was associated with pericardial effusion. No pathological changes in the ECG compared to the initial changes were recorded in 14.9% of patients, and the diagnosis of myocarditis was confirmed based on the clinical picture and changes in the laboratory parameters and/or MRI and/or myocardial biopsy data.

The results of EchoCG were not described in 53 cases. Despite the severe course of ICT-associated myocarditis, only 50 patients (26.2%) among the presented clinical cases had a violation of the contractile ability of the left ventricular myocardium with a decrease in the ejection fraction of <40%. In 12 cases (6%), the ejection fraction was moderately reduced (40–49%). In addition, in 129 cases (64.2%), a preserved ejection fraction was diagnosed.

The assessment of myocardial damage markers (troponin and creatine kinase (CK), CK-MB), markers of heart failure development (brain natriuretic peptide (BNP) or N-terminal probrain natriuretic peptide (NT-proBNP)) was not presented in all of the articles. The troponin level was presented in 197 clinical cases, CK-MB and/or CK in 174, and BNP or NT-proBNP in only 80. Among them, an increased level of troponin was detected in 93.9% of cases (n = 185), an increase in CK and/or CK-MB was noted in 93.7% of patients (n = 163), and an increase in BNP or NT-proBNP in 77.5% (n = 185).

Data on cardiac MRI were presented in 75 clinical cases, but in many studies, there was no clear information that would reliably confirm the diagnosis of myocarditis. In four cases, there were indications that MRI did not reveal any signs of the current inflammatory processes in the myocardium, but the diagnosis of myocarditis was confirmed during morphological examination, taking into account the clinical picture, laboratory changes, and instrumental studies.

Morphological examination of the myocardium was performed in 61 patients, and in other scientific works, there is no information about the biopsy performed. In eight cases, the inflammatory process in the myocardium was confirmed by autopsy examination. In most cases, a lymphocytic infiltrate was described (which was positive for T-cell markers such as CD4 and CD8) with positive CD68 macrophages.

In 103 cases (42.2%), in addition to myocardial damage, patients were diagnosed with myositis and/or myasthenia. Mortality in the group of patients with ICI-associated myocarditis and myositis/myasthenia in our analysis was 29.4%, which exceeded the mortality rates in the group of patients without myositis and/or myasthenia (23.9%).

The first line of therapy typically includes glucocorticosteroids (GCs), which are widely used to treat autoimmune diseases, cytokine storms, and can be effective in the managing of relief complications caused by checkpoint inhibitors. In 10 articles, there was no detailed description of the therapy performed. In 97% of cases, the patients received therapy with prednisolone or methylprednisolone. As a second line of therapy, 22.6% of the patients received intravenous immunoglobulin (IVIG) infusions and 3.4% received antithymocyte immunoglobulin infusions. Immunosuppressive drugs, such as mycophenolate mofetil (11.5%), infliximab (6.8%), tofacitinib (5.1%), abatacept (2.6%), rituximab (1.4%), methotrexate (0.9%), cyclophosphamide (0.9%), tocilizumab (0.4%), alemtuzumab (0.4%), ruxolitinib (0.4%), and tacrolimus (0.4%), were less frequently prescribed. Plasmapheresis was performed in 9% of cases, ECMO (extracorporeal membrane oxygenation) in 1.7%, temporary or permanent cardio stimulator was mentioned in 19.7% of cases, and intra-aortic balloon counterpulsation was performed in 2.6% of cases for life-saving indications.

### 3.2. Deceased Patients

Despite undergoing therapy, the lethality rate from ICI-associated myocarditis was 26.2%. In the subgroup of patients who died as a result of ICI-associated myocarditis, 42 (65.6%) were male, with a median age of 69.5 (IQR, 63–77) years. The age at which the first symptoms of the disease developed in women was 66 (IQR, 57–70.5) years, which was lower than that of men at 73 (IQR, 66–79.75) years, as well as among all the patients included in the analysis.

In the subgroup of deceased patients, checkpoint inhibitors were prescribed to treat the following oncological diseases: melanoma in 17 cases (26.6%), lung cancer in 12 (18.7%), thymoma in 8 (12.5%), renal cell carcinoma in 5 (7.8%), urothelial carcinoma in 3 (4.7%), and other locations each in 1–2 described cases—total in 19 (29.7%).

Upon evaluating the therapy being administered, it was found that fatal outcomes occurred in 82.8% of ICI-associated myocarditis cases against the background of monotherapy with drugs from the anti-PD-1 group (primarily pembrolizumab (32.8%) and nivolumab (26.6%)). Other treatment options were significantly less common; combination therapy with anti-CTLA4 and anti-PD-1 was used in 1.6% of cases, monotherapy with anti-CTLA-4 in 3.1%, and anti-PD-L1 in 3.1%. Thus, it can be assumed that there is the highest risk of an unfavorable outcome in the group receiving anti-PD-1 therapy, especially with pembrolizumab and nivolumab. However, it cannot be ignored that the obtained results are due to the higher frequency of the use of these drugs in real practice.

The median time from the beginning of ICI therapy to the development of the first symptoms from the cardiovascular system was 15.5 days (IQR, 12–28.5), and the median day of death after the first use of ICI was 31 days (IQR, 19–54).

In one case, the patient’s complaints were not described. Among the remaining 63 cases, cardiac manifestations at the time of onset of the disease were registered in 50.8% of patients, neuromuscular in 30.2%, and other manifestations in 12.7%. In 6.3% of cases, the disease started asymptomatically.

In seven of the analyzed cases, the results of the ECG were not described, so the frequency of occurrence of various types of rhythm and conduction disorders was evaluated in 57 patients. In 96.5% of cases, new changes in the ECG were registered compared to the data before the start of ICI use. As with all the analyzed patients, conduction disturbances predominated, including atrioventricular block of different degrees and bundle branch blocks. In 40.4% of cases, a third-degree atrioventricular block was registered, but less frequently (first degree in 7% and second degree in 3.5% of cases). Left bundle branch block was detected in 8.8% of cases, and right bundle branch block in 28.1%. Among the supraventricular arrhythmia types, sinus tachycardia (12.3%), atrial fibrillation (7%), and atrial flutter (1.8%) were described. Ventricular tachycardia developed in 17.5% of patients. Premature atrial contractions (1.8%) and premature ventricular contractions (7%) were infrequently registered. ST segment elevation in 22.7% of cases, ST segment depression in 3.5%, and T wave inversion in 5.3% were also described. In one case (1.8%), the appearance of a Q wave was observed. Low voltages in several leads were described in 3.5% of cases. Pathological changes in the ECG compared to the initial data were not detected in 3.5% of patients.

According to EchoCG data, which was described in 51 cases, decreased ejection fraction was detected in 29.4% of cases, moderately decreased in 17.6%, and preserved in 52.9%. Laboratory markers of myocardial damage were not mentioned in any of the analyzed scientific works. The level of CK and/or CK-MB was evaluated in 50 cases and was elevated in 94%. The level of troponins was evaluated in 55 cases and was elevated in 92.7%. The evaluation of BNP or NT-proBNP was indicated in 20 articles, and its increase was registered in 85%.

Cardiac MRI was performed in 20.3% of cases and endomyocardial biopsy in 29.7% (of which autopsy was performed in 12.5%).

Myositis and/or myasthenia developed in 46.9% of patients.

Therapy for ICI-associated myocarditis was described in 60 patients. Glucocorticoid therapy was performed in almost all cases (98.3%). IVIG was used in 26.7% of cases, and antithymocyte immunoglobulin infusions in 5%. The following drugs were also mentioned for treatment: mycophenolate mofetil (5%), infliximab (18.3%), tofacitinib (6.3%), rituximab (1.7%) and cyclophosphamide (1.7%). Plasmapheresis was performed in 10% of cases, ECMO in 1.7%, and in 28.1% of cases, a cardio stimulator was required.

## 4. Discussion

Cardiotoxicity, although a rare side effect of ICI inhibitors, is concerning due to its high mortality rate and early development that often occurs soon after the start of treatment. While most cases occur soon after the start of ICI treatment—with a median of 21 days (IQR, 14–38.3)—there are several cases that manifest significantly later. Based on this information, clinicians should pay particular vigilance to cardiac toxicity in patients soon after the start of treatment. For example, according to our analysis, nearly one third of patients developed myocarditis after the first use of ICI. Myocarditis develops more quickly in the Atezolizumab group—14 days (IQR, 4–30)—which is also confirmed by Shalata W et al. [197], but with the smaller number of patients in this subgroup, this issue should be studied in prospective studies. However, it should be remembered that the risk of cardiotoxicity never completely disappears, and patients may need to be monitored for suspicious symptoms even years after the start of ICI treatment.

Myocarditis has the highest mortality rate among all types of ICI-induced cardiotoxicity. However, if they do not progress to a fatal outcome, these side effects are often quickly treated with steroid treatment [194]. According to our data, the mortality rate from ICI-associated myocarditis was 26.2%. This index is at the lower end of those mentioned in earlier scientific works. Perhaps this is due to the large number of studies from 2021 and 2022 included in our review, when doctors were already alert to the possibility of ICI-associated myocarditis and, in cases of asymptomatic increase in cardiac-specific markers, stopped using the drug, conducted further testing of the patients, and provided necessary therapy, primarily with high doses of GCs.

The clinical presentation of ICI-associated myocarditis can vary from asymptomatic elevation of cardiac biomarkers (which corresponds to grade 1 severity) to severe decompensation with the involvement of target organs. Fulminant myocarditis is the most concerning, and cases are often reported in the literature. In these patients, myocarditis is often accompanied by serious arrhythmias and conduction disturbances, such as complete atrioventricular block and ventricular tachycardia. It should be noted that if patients with cardiac symptoms also have symptoms of another irAE, the possibility of myocarditis associated with ICI increases [198]. Myocarditis is most often associated with myasthenia and myositis in such cases. This is likely due to the antigenic relationship between cardiomyocytes and skeletal muscles.

There are several scientific works that describe the overlap of ICI-associated myocarditis with myositis and/or myasthenia, mainly in the form of case reports or small case series; the frequency of concomitant myositis and myocarditis is 30–40% and myasthenia and myocarditis is 10% [32,199,200]. According to our data, myositis and/or myasthenia developed in 46.9% of patients, and in 30.3% of cases, complaints related to the nervous system or muscle involvement were the first manifestation of the disease that prompted patients to ask for medical help. Given the relatively high incidence, patients with immune-mediated myositis and/or myocarditis should be promptly screened for myasthenia, considering the risk of myasthenic crisis and adverse outcomes, such as respiratory paralysis and death [40], through the analysis of repetitive nerve stimulation and analysis of autoantibodies against the acetylcholine receptor. However, they are not detected in all patients (only approximately 60%) [40]. The mortality rate in the group of patients with ICI-associated myocarditis and myositis/myasthenia, according to our analysis, was 29.4%, which exceeded the mortality rate in the group of patients without myositis and/or myasthenia, which was 23.9%. These indexes are lower than those previously described; Pathak R et al. found significant in-hospital mortality in cases of cross-involvement, with mortality rates approaching 60% [199]. In this situation, we can also suggest that doctors were more vigilant and used treatment methods that have previously shown high effectiveness, such as the addition of steroids, IVIG and plasmapheresis of other immunosuppressants such as tacrolimus, infliximab, mycophenolate mofetil, or antithymocyte globulin.

The diagnostics of ICI-associated myocarditis involves the use of ECG, evaluation of troponin, BNP or NT-proBNP, and echocardiograms. Depending on the indications, stress tests, coronary angiography, and cardiac MRI may be performed [16].

Overall, the diagnostics strategy of ICI-associated myocarditis is potentially more complex than the diagnostics strategy for myocarditis of another etiology. Its impact on patient treatment outcomes is significant, considering its consequences. Under-diagnosis may lead to the absence or delay of corticosteroid therapy and major adverse cardiac events (MACE). Overdiagnosis may lead to the discontinuation of ICI treatment and progression of cancer. Bonaca et al. proposed a unified definition of cancer therapy-related myocarditis [201]. According to this definition, we can speak of definite myocarditis in three situations. The first variant is associated with signs of myocarditis that are found during histological examination (myocardial biopsy or autopsy). The second variant is associated with typical changes in cardiac MRI, along with an increase in the markers of myocardial necrosis in the blood or in combination with typical changes on an ECG. In the third variant, abnormal movement of the myocardial walls is established during echocardiography, which cannot be explained by other causes in combination with clinical symptoms and an increase in markers of myocardial necrosis and changes on an ECG in the absence of changes during angiography (or other techniques that confirm the absence of obstructive coronary artery disease).

Probable myocarditis is confirmed in the following scenarios. For first variant, typical changes in cardiac MRI are not accompanied by a typical clinical picture, an increase in the markers of myocardial necrosis, or changes on an ECG. Usually, such changes are detected accidentally or as part of clinical research procedures. For the second variant, non-specific but suspicious myocarditis changes are detected on cardiac MRI, in combination with a typical clinical picture and/or an increase in the markers of myocardial necrosis and/or changes on an ECG. In the third variant, abnormal movement of the myocardial walls is established during EchoCG in combination with clinical symptoms of myocarditis and an increase in the markers of myocardial necrosis in the blood and changes on an ECG. In the fourth variant, suspicious changes are detected on PET-CT.

Possible myocarditis is established in the following variants. For the first variant, non-specific but suspicious changes in myocarditis are detected on cardiac MRI without a typical clinical picture, in addition to an increase in the markers of necrosis, or changes on an ECG. The second variant is the detection of abnormal movement of the myocardial walls during echoCG, in combination with changes on an ECG or clinical symptoms of myocarditis. The third variant is an increase in the markers of necrosis compared to the baseline level in combination, with clinical symptoms of myocarditis or typical changes on an ECG.

However, these recommendations are primarily intended for use in clinical trials [202]. Another definition of immune checkpoint inhibitor myocarditis has recently been proposed by the International Cardio-Oncology Society [203]. According to this definition, the diagnosis of myocarditis can be established in the following situations. The first variant is histological confirmation based on myocardial biopsy or autopsy data. The second variant is a clinically confirmed diagnosis in combination with elevated cardiac troponins and typical changes in cardiac MRI. The third variant is a clinically confirmed diagnosis in combination with elevated cardiac troponins with two of the following listed features: typical clinical symptoms, lower extremity edema, ventricular arrhythmia or new conduction disturbance, decreased ejection fraction, other irAEs and suspicious myocarditis changes in cardiac MRI.

Criteria for complete recovery and ongoing recovery have also been developed [204].

Complete recovery is associated with patients with a complete resolution of acute symptoms, normalization of biomarkers, and the recovery of ejection fraction after the discontinuation of immunosuppression. Cardiac MRI may still show late gadolinium enhancement or elevated T1 due to fibrosis, but any suggestion of acute oedema should be absent.

MRI is the gold standard for the non-invasive diagnostics of myocarditis. According to our analysis, cardiac MRI was performed in only 75 cases (30.7%), but in many scientific works, specific data confirming or excluding myocarditis according to the existing criteria [205] are absent. Myocardial biopsy as part of the diagnostics of ICI-associated myocarditis cannot always be performed due to the severity of the patient’s condition, patient refusal, or, conversely, the absence of serious clinical symptoms. According to our data, it was performed in 61 cases (25%). In addition, it cannot be excluded that these diagnostic methods were not needed or were uninformative due to the positive effect of the conducted GC therapy.

Cardiac troponins are sensitive and are specific markers of myocardial damage. In patients with clinical manifestations that indicate acute coronary syndrome, coronary angiography may be required for the purpose of differential diagnosis. In a previous case–control study, troponin was elevated in 94% of myocarditis cases associated with ICI, while BNP or NT-proBNP was elevated in 66% of cases [30]. In our research, troponin levels were elevated in 93.9% of cases, and CK and/or CK-MB levels were elevated in 93.7% of cases. Most patients that were suspected of having ACS underwent coronary angiography. No obstructive changes in the coronary arteries were detected in any of the cases. However, it should be noted that in a small percentage of cases, the markers of myocardial damage may be normal. NT-proBNP was elevated in 77.5% of cases according to our data, but its elevation is also possible in many cancer patients due to inflammation associated with cancer; therefore, the result is non-specific.

EchoCG allows us to evaluate the structural and functional changes provoked by myocarditis. In our research, the majority of patients—64.2%—showed preserved ejection fraction, even in the group of deceased patients (52.9%). However, it is possible to suggest that death occurred before the development of cardiac dysfunction in these cases. Ejection fraction of less than 40% was detected in only 26.2% of all patients and 29.4% of deceased patients according to our analysis. However, EchoCG allows us to evaluate the dynamics of functional parameters against the background of the conducted therapy. In addition, EchoCG can show changes in the heart chambers, geometry, or regional wall motion abnormalities. At the same time, the preservation of normal heart size may indicate an acute process, while remodeling and dilation indicate a chronic process.

ICI-associated myocarditis in most cases is accompanied by changes on an ECG. According to our data, ECG changes were detected in 85.1% of patients, which were not present before the use of ICI. Conduction disorders predominated, with complete atrioventricular block being registered in 25.4% and complete right bundle branch block in 18.4% of cases and among arrhythmias, the most common types of disturbances were sinus tachycardia (12%) and ventricular tachycardia (13%). The presence of a high frequency of serious conduction disturbances may be associated with T-cell-mediated cytotoxicity, which affects the cardiac conduction system. Histological studies have shown that ICI-mediated cardiomyocyte necrosis is characterized by the infiltration of CD4+ and CD8+ T-cells and is similar to the development of acute cardiac rejection after transplantation [205,206]. Lymphocytic infiltration may involve the sinoatrial and atrioventricular nodes [207].

The main treatment for ICI-associated cardiotoxicity involves the use of high doses of corticosteroids (1–2 mg/kg) [23,24]. As a rule, the duration of glucocorticoid use after the resolution of symptoms is at least 4–6 weeks [208]. If the effectiveness is insufficient, other drugs may be used, such as mycophenolate mofetil, infliximab, IV immunoglobulin, or antithymocyte globulin, but their use is not associated with the recommendations for the diagnostics and treatment of ICI-associated myocarditis [23]. According to our data, the most common treatment options were as follows: 97% of patients received therapy with prednisolone or methylprednisolone, 22.6% received IVIG infusions, 11.5% received mycophenolate mofetil, and 6.8% received infliximab.

Regarding the further use of ICI after the development of ICI-associated myocarditis, some scientific works recommend temporarily discontinuing ICI use even after minor cardiotoxicity; if there is evidence of severe toxicity, it should be permanently discontinued [209]. Among the clinical cases analyzed by us, it was only mentioned in two cases that ICI use was resumed later. However, ICI has a long half-life in the body, so discontinuing treatment will not lead to an immediate change in the biological activity of the drug [210]; this is a justification for the prolonged use of GCs.

How can we detect ICI-associated myocarditis in its early stage? In order to detect possible cardiotoxicity early, several studies recommend routine cardiac examination (testing for BNP or NT-proBNP, cardiac troponin, and ECG) before starting ICI treatment and during the first 1–4 cycles or up to 12 weeks of treatment [210,211,212,213]. The latest recommendations for cardio-oncology suggest the following scenario for detecting ICI-associated cardiotoxicity [213,214]: initial evaluation of ECG, BNP or NT-proBNP and cardiac troponin, followed by a reanalysis of ECG and cardiac troponins before the 2nd, 3rd, and 4th dose of the medication. If these tests do not show any changes, repeated examination should be performed after every three doses of the medication.

## 5. Conclusions

Immune checkpoint inhibitors can cause various side effects related to pathological autoimmunity. One of the rare side effects is cardiotoxicity manifested as myocarditis. Although these side effects are rare, they have a high mortality rate. When using ICI, particular attention should be paid to the early detection of possible cardiotoxicity, including analyzing the state and functions of the myocardium before treatment and in dynamics. The early detection of ICI-associated myocarditis and timely discontinuation of ICI use may lead to a more favorable outcome in such cases. The main proven treatment method is the use of high doses of steroids.

## Figures and Tables

**Table 1 diagnostics-13-01243-t001:** ICIs in oncology.

Blocked Checkpoint Protein	International Non-Proprietary Name	First Approvals *	Indications for Use (as of 2023) *
CTLA-4	Ipilimumab	FDA, 2011 EMA, 2011	Melanoma, renal cell carcinoma, colorectal cancer, hepatocellular carcinoma, non-small cell lung cancer, malignant pleural mesothelioma, esophageal cancer
	Tremelimumab	FDA, 2022	Hepatocellular carcinoma
PD-1	Cadonilimab [5]	CNMPA, 2022	Cervical cancer
	Camrelizumab [6,7]	CNMPA, 2019	Hodgkin’s lymphoma, hepatocellular carcinoma, non-small cell lung cancer, esophageal cancer
	Cemiplimab	FDA, 2018 EMA, 2019	Cutaneous squamous cell carcinoma, basal cell carcinoma, non-small cell lung cancer
	Nivolumab	FDA, 2014 EMA, 2015	Melanoma, non-small cell lung cancer, malignant pleural mesothelioma, renal cell carcinoma, classical Hodgkin lymphoma, squamous cell carcinoma of the head and neck, urothelial carcinoma, colorectal cancer, hepatocellular carcinoma, esophageal cancer, gastric cancer, gastroesophageal junction cancer, esophageal adenocarcinoma
	Pembrolizumab	FDA, 2014 EMA, 2015	Melanoma, non-small cell lung cancer, head and neck squamous cell cancer, classical Hodgkin lymphoma, primary mediastinal large b-cell lymphoma, urothelial carcinoma, microsatellite instability-high or mismatch repair deficient cancer, microsatellite instability-high or mismatch repair deficient colorectal cancer, gastric cancer, esophageal cancer, cervical cancer, hepatocellular carcinoma, Merkel cell carcinoma, renal cell carcinoma, endometrial carcinoma, tumor mutational burden-high cancer, cutaneous squamous cell carcinoma, triple-negative breast cancer, adult classical Hodgkin lymphoma and adult primary mediastinal large b-cell lymphoma
	Sintilimab [8]	CNMPA, 2018 EMA, 2020	Melanoma, esophageal cancer, nasopharyngeal cancer, small cell lung cancer, soft tissue sarcoma, peripheral T-cell lymphoma
	Tislelizumab [9]	CNMPA, 2019 EMA, 2020	Non-small cell lung cancer, esophageal squamous cell carcinoma
	Toripalimab [10]	CNMPA, 2018 EMA, 2022	Melanoma, esophageal squamous cell carcinoma, nasopharyngeal cancer, urothelial carcinoma
PD-L1	Atezolizumab	FDA, 2016 EMA, 2017	Urothelial carcinoma, non-small cell lung cancer, non-small cell lung cancer, hepatocellular carcinoma, melanoma, alveolar soft part sarcoma
	Durvalumab	FDA, 2017 EMA, 2018	Non-small cell lung cancer, small cell lung cancer, biliary tract cancer, hepatocellular carcinoma
LAG-3	Relatlimab	FDA, 2022 EMA, 2022	Melanoma (in combination with nivolumab)

CNMPA: China National Medical Products Administration; FDA: Food and Drug Administration; EMA: European Medicines Agency. *—information from sites: IUPHAR/BPS Guide to PHARMACOLOGY, https://www.guidetopharmacology.org; Drugs@FDA: FDA-Approved Drugs, https://www.accessdata.fda.gov/scripts/cder/daf/index.cfm; EMA, Medicine, https://www.ema.europa.eu/en/medicines/field_ema_web_categories%253Aname_field/Human (accessed on 13 January 2023).

**Table 2 diagnostics-13-01243-t002:** IrAE grading system.

The Organ(s)	Grade 1	Grade 2	Grade 3	Grade 4
Acute kidney injury (creatinine increased)	<1.5× baseline or upper limit of normal	1.5–3.0× baseline or upper limit of normal	3.0–6.0× baseline or upper limit of normal or >3.0× baseline, but dialysis not indicated	>6.0× baseline or upper limit of normal or Dialysis indicated
Arthritis	Mild pain with inflammation, redness, or joint swelling	Moderate pain with inflammation, redness, or joint swelling Restriction of instrumental ADL	Severe pain with inflammation, redness, or joint swelling Irreversible joint injury Restriction of self-care	
Colitis	Asymptomatic <4 bowel movements in 24 h	Abdominal pain Mucus or blood in feces 4–6 bowel movements in 24 h	Severe abdominal pain Peritoneal signs >7 bowel movements in 24 h	Life-threatening consequences
Liver damage (ALT increased)	<3.0× upper limit of normal Asymptomatic	3.0–5.0× upper limit of normal Asymptomatic	5.0–20.0× upper limit of normal Symptomatic Compensated cirrhosis	20.0× upper limit of normal Liver function decompensation (ascites, dysfunction of hemostasis, toxic damage to the central nervous system, coma)
Pituitary lesion	Asymptomatic Mild symptoms	Moderate symptoms limiting age-appropriate instrumental ADL	Severe or medically significant limiting self-care ADL	Life-threatening consequences
Skin rash	Less than 10% of body surface area affected	10–30% of body surface area affected	More than 30% of body surface area affectedLife-threatening manifestations Generalized exfoliation, skin ulcers	
Myasthenia gravis	No or mild manifestations	Moderate manifestations Limiting age-appropriate instrumental ADL	Severe manifestations Limiting self-care ADL	Life-threatening consequences, dysfunction of the respiratory muscles
Myositis	Pain	Pain associated with mild or moderate weakness; Limitation of instrumental ADL	Pain with severe weakness; limitation of self-service	Life-threatening consequences; urgent intervention indicated
Myocarditis	Asymptomatic Cardiac enzyme elevation or abnormal electrocardiogram (EСG)	Symptoms with moderate activity	Manifestations at rest and with minimal exertion	Life-threatening consequences
Arrhythmias	Asymptomatic, intervention not indicated	Non-urgent medical intervention indicated	Symptomatic, urgent intervention indicated (pacemaker, ablation, etc.)	Life-threatening consequences; embolus requiring urgent intervention
Atrioventricular block complete		Non-emergency intervention indicated	Partially medication controlled, or controlled with device (e.g., pacemaker)	Life-threatening consequences; urgent intervention indicated
Conduction disorder	Intervention not needed	Non-emergency medical intervention	Emergency intervention	Life-threatening consequences
Heart failure	Asymptomatic with laboratory (e.g., BNP (brain natriuretic peptide)) or cardiac imaging changes	Symptoms with moderate activity or exertion	Symptoms at rest or with minimal activity or exertion	Emergency intervention indicated
Ventricular tachycardia		Non-urgent medical intervention indicated	Symptomatic, urgent intervention indicated	Life-threatening consequences; hemodynamic compromise
Pericardial effusion		Asymptomatic	Symptomatic	Emergency intervention indicated
Pericarditis	Asymptomatic	Symptomatic, mild or moderate manifestations	Physiologic consequences, severe manifestations	Life-threatening consequence
Pneumonitis	Asymptomatic Only one lobe of the lung is affected or less than 25% of the lung	Symptomatic (dyspnea, cough or chest pain) Two or more lobes of the lung are affected or 25–50% of the lung	Severe symptoms (new or worsening hypoxia) All lung lobes of the lung are affected or >50% of the lung Oxygen therapy	Life-threatening respiratory compromise

ADL: activity of daily living.

**Table 3 diagnostics-13-01243-t003:** Demographic and clinical characteristics of individual case reports.

Characteristics	Patients N (%)
Median age (year)	67 (IQR, 60–74)
Male sex (%)	143 (58.6%)
Female sex (%)	99 (40.6%)
Gender not specified (%)	2 (0.8%)
Types of ICI	
Pembrolizumab	75 (30.7%)
Nivolumab	49 (20.1%)
Nivolumab plus ipilimumab	41 (16.8%)
Sintilimab	20 (9.3%)
Camrelizumab	15 (6.1%)
Durvalumab	8 (3.3%)
Toripalimab	7 (2.9%)
PD-L1 inhibitor (investigational)	6 (2.6%)
Atezolizumab	5 (2%)
Tislelizumab	5 (2%)
Others	13 (5.3%)

**Table 4 diagnostics-13-01243-t004:** Days until presentation of ICI-associated cardiotoxicity, organized by treatment.

ICI	Sintilimab	Nivolumab	Pembrolizumab	Ipilimumab + Nivolumab	Durvalumab	Atezolizumab	Toripalimab	Tislelizumab	Camrelizumab
Number of cases	20	49	75	41	8	5	7	5	15
Median days until manifestation	21	23	26	19	34	14	20	20	25

**Table 5 diagnostics-13-01243-t005:** Electrocardiographic findings.

Findings	N (%)
Sinus arrhythmia	
Sinus tachycardia	24 (12%)
Sinus bradycardia	5 (2.5%)
Sick sinus syndrome	2 (1%)
Atrial arrhythmia	
Atrial extrasystole	3 (1.5%)
Atrial flutter	3 (1.5%)
Atrial fibrillation	16 (8%)
Ventricular arrhythmia	
Ventricular extrasystole	11 (5.5%)
Ventricular tachycardia	26 (13%)
Atrioventricular block	
1-degree atrioventricular block	8 (4%)
2-degree atrioventricular block	12 (6%)
3-degree atrioventricular block	51 (25.4%)
Bundle branch block	
Left bundle branch block	12 (6%)
Right bundle branch block	37 (18.4%)
ST segment changes	
ST segment elevation	33 (6.4%)
ST segment depression	12 (6%)
T wave inversion	21 (10.4%)
Non-classified	
New Q wave	1 (0.5%)
Low voltage	8 (4%)
No changes	30 (14.9%)

## Data Availability

Not applicable.

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
