# Peer review of "Immune Checkpoint Inhibitors-Related Myocarditis: A Review of Reported Clinical Cases"

_diagnostics, 2023, doi:10.3390/diagnostics13071243_

Round 1

Reviewer 1 Report

The paper is thorough and detailed. The time and patience needed for such an emormous data collection and analysis is invaluable. I commend You for the work done here.

Regardless, there are few things which need correction, even though overall they do not deminish the worth of the work;

  • The min-max values of certain parameters should be written as [min - max] not [min;max];
  • Table 1 lacks proper citations;
  • The data in table 3 are mostly already presented in the main text, therefore it should be removed or limited to data not detailed in the main text only;
  • The data in table 5 are already presented in the main text, either leave the table or the main text information;

Additional changes are marked in attached file.

Author Response

I thank the reviewer for analyzing the article and providing feedback and suggestions that helped us to improve the presentation of the data. I took into account the comments and suggestions in the corrected version of the manuscript.

  1. The min-max values of certain parameters should be written as [min - max] not [min;max];

Answer: For the quantitative variables, data show median and interquartile range (IQR, Q1-Q3). Formatting for writing the interquartile range has been corrected throughout the text, and corresponding information has been added to the "Materials and Methods".

  1. Table 1 lacks proper citations;

Answer: The information in this table was independently collected from guidelines for the use of medicinal products from the following websites:

https://www.guidetopharmacology.org/

Drugs@FDA: FDA-Approved Drugs, https://www.accessdata.fda.gov/scripts/cder/daf/index.cfm, 

EMA,Medicine https://www.ema.europa.eu/en/medicines/field_ema_web_categories%253Aname_field/Human

A footnote has been added to Table 1. Additionally, references to literature sources that were not previously cited have been added.

  1. The data in table 3 are mostly already presented in the main text, therefore it should be removed or limited to data not detailed in the main text only;

Answer: The main text and information in the table have been corrected to minimize duplicate data.

  1. The data in table 5 are already presented in the main text, either leave the table or the main text information;

Answer: The main text has been corrected to include only the necessary information for emphasis (only the most common variations). In addition, based on this and the previous recommendation, the table describing the characteristics of deceased patients has been removed, as it also contained information that was already in the text.

  1. Additional changes are marked in attached file.

Answer: Thank you very much for the recommended corrections - they have all been made in the text.

Reviewer 2 Report

I appreciate the authors collected the case reports under the title of “Immune Checkpoint Inhibitors-Related Myocarditis: A Review Of Reported Clinical Cases”

1.      Authors have written the table title as Classification of ICIs and examples of the main registered drugs in use.  ICI treatment always associated with cancer, the table title could be changed as  ICI drugs associated with cancer.

2.      Authors suggested to re write the statement for better clarification instead of

“When inhibitory receptors, such as CTLA-4 and PD-1, which are expressed on T lymphocytes, bind to their corresponding ligands on tumor cells, such as PD-L1, the cellular

immune response is "turned off”

Both CTLA-4 and PD-1 and its partner of PD-L1, Drugs inhibits the checkpoint protein CTLA-4. Other immune checkpoint drugs target the checkpoint proteins PD-1 and PD-L1, which are partners in the immune system. Certain tumors inhibit the T cell reaction by overproducing PD-L1.

3.Authors conducted the ECG report to the patients were under immune therapy.

 But, in general, depends on the severity and stage of the cancer the concentration and dose limit would be preferred by the physicians. If authors provide the details of disease stage and dose limits, it could be easy to correlate with cardiac function variations and impacts of the selected drugs.

4.      Line number 233, hors have mentioned that the biomarkers of myocardial disease “myocardial damage markers was not presented in all articles”.

There are articles available. But the authors referred articles did not covered those details. So, authors advised to rewrite the statement that the “ myocardial damage markers were not presented from selected articles or referred articles.

Author Response

Thank you very much for your careful attention to the article and the feedback you provided. All of your comments have been taken into account, corresponding edits have been made to the text, and they are described in detail below.

  1. Authors have written the table title as Classification of ICIs and examples of the main registered drugs in use.  ICI treatment always associated with cancer, the table title could be changed as  ICI drugs associated with cancer.

Answer: The title of the table has been changed.

  1. Authors suggested to re write the statement for better clarification instead of “When inhibitory receptors, such as CTLA-4 and PD-1, which are expressed on T lymphocytes, bind to their corresponding ligands on tumor cells, such as PD-L1, the cellular immune response is "turned off”

Both CTLA-4 and PD-1 and its partner of PD-L1, Drugs inhibits the checkpoint protein CTLA-4. Other immune checkpoint drugs target the checkpoint proteins PD-1 and PD-L1, which are partners in the immune system. Certain tumors inhibit the T cell reaction by overproducing PD-L1.

 Answer: The text has been changed.

3.Authors conducted the ECG report to the patients were under immune therapy.

 But, in general, depends on the severity and stage of the cancer the concentration and dose limit would be preferred by the physicians. If authors provide the details of disease stage and dose limits, it could be easy to correlate with cardiac function variations and impacts of the selected drugs.

Answer: The research is retrospective and analyzes data described by other authors, that is why we had limited information. For example, in many research works, there is no indication of the stage of the cancer, only the type of cancer is mentioned. Among research works where the case history is described in more detail, it is indicated that patients had an advanced stage - with detached metastases. Therefore, it is not possible to correlate ECG changes and the stage of the disease. Additionally, not all research works indicate the dose of the drug or the number of cycles; often only the number of days from the start of ICI use to the onset of the first myocarditis symptoms is mentioned. All of this clearly indicates the necessity for prospective research works to evaluate ICI toxicity, which could establish more accurate correlations between myocarditis symptoms and therapy.

  1. Line number 233, hors have mentioned that the biomarkers of myocardial disease “myocardial damage markers was not presented in all articles”. There are articles available. But the authors referred articles did not covered those details. So, authors advised to rewrite the statement that the “myocardial damage markers were not presented from selected articles or referred articles.

Answer: Thank you very much for the feedback! Of course, I expressed my thought incorrectly that not all analyzed articles contained the necessary information. Moreover, later in the text, I indicate in how many cases the necessary data was presented. The text has been changed.